# Tumor Segmentation in Breast Ultrasound Image by Means of Res Path Combined with Dense Connection Neural Network

**DOI:** 10.3390/diagnostics11091565

**Published:** 2021-08-28

**Authors:** Kailuo Yu, Sheng Chen, Yanghuai Chen

**Affiliations:** Department of Automatic Control, School of Optical-Electrical and Computer Engineering, University of Shanghai for Science and Technology, 516 Jungong Road, Shanghai 200093, China; yukl9705@163.com (K.Y.); yhuaichen123@163.com (Y.C.)

**Keywords:** tumor segmentation, breast ultrasound, Res Path

## Abstract

Over the past few years, researchers have demonstrated the possibilities to use the Computer-Aided Diagnosis (CAD) to provide a preliminary diagnosis. Recently, it is also becoming increasingly common for doctors and computer practitioners to collaborate on developing CAD. Since the early diagnosis of breast cancer is the most critical step, a precise segmentation of breast tumor with accurate edge and shape is vital for accurate diagnoses and reduction in the patients’ pain. In view of the deficient accuracy of existing method, we proposed a novel method based on U-Net to improve the tumor segmentation accuracy in breast ultrasound images. First, Res Path was introduced into the U-Net to reduce the difference between the feature maps of the encoder and decoder. Then, a new connection, dense block from the input of the feature maps in the encoding-to-decoding section, was added to reduce the feature information loss and alleviate the vanishing gradient problem. A breast ultrasound database, which contains 538 tumor images, from Xinhua Hospital in Shanghai and marked by two professional doctors was used to train and test models. We, using ten-fold cross-validation method, compared the U-Net, U-Net with Res Path, and the proposed method to verify the improvements. The results demonstrated an overall improvement by the proposed approach when compared with the other in terms of true-positive rate, false-positive rate, Hausdorff distance indices, Jaccard similarity, and Dice coefficients.

## 1. Introduction

Breast cancer is the most common cancer in women and second leading cause of cancer death [1]. More than eight percent of women are expected to develop it in their lifetime [2]. Because of the high incidence and gruesomeness of breast cancer, medical researchers have paid close attention to it. Accurate and early diagnosis is the key to preventing it [3].

Biopsy is the gold standard for diagnosing breast cancer, and its pathological results often serve as the conclusive result of breast cancer diagnosis [4]. However, it requires more manpower and material resources and can damage the normal physiological tissues of patients, causing both physical pain and psychological damage. Therefore, a biopsy is not the best choice for patients. With the rapid advances in computer technology, medical imaging technology has emerged and has inspired many new breast cancer diagnosis methods. Recently, ultrasound (US) is commonly used as a complementary method for breast cancer detection due to its versatility, safety, and high sensitivity [5].

However, compared with other commonly used techniques, such as mammography, breast cancer diagnosis based on ultrasonic images often requires experienced and well-trained doctors. This is because the images sometimes contain speckle noise and are of lower resolution, which could lead to high subjectivity and uncertain results. Consequently, breast cancer diagnosis based on ultrasonic imaging can be easily affected by the peculiarities of individual cases and the level of experience of medical personnel; these factors could lead to misdiagnosis and other adverse consequences.

Fortunately, CAD has greatly developed in recent years and could be beneficial to help radiologists in the US-based detection of breast cancer, minimizing the effect of the operator-dependent nature of US imaging. Different studies have investigated the influence of CAD on diagnostics and showed that CAD is an important tool to improve the diagnostic sensitivity and specificity. It also has been exploited by relevant medical personnel to diagnose and discriminate tumors for improved diagnostic accuracy. The location, shape, and size of tumors can be determined by CAD programs. Thus, the ability to locate the lesion is the most critical for a CAD program when assisting the doctor in mapping the region of the tumor. During this process, segmenting the tumor region from the breast US images automatically with high accuracy is of primary importance, and a high sensitivity and specificity are expected.

With the improvement of computer technology, deep-learning approaches have surpassed traditional ones and become the predominant choices for breast ultrasound image segmentation [6]. Several deep-learning models have been proposed, such as AlexNet [7], VGG [8], GoogLeNet [9], Residual network [10], and DenseNet [11], and they all have excellent performance in numerous areas. But the Fully Convolutional Networks (FCN) [12] provide superior performance when compared with other deep-learning models with respect to semantic medical image segmentation. In subsequent studies, many of them are based on the FCN architectures, and therefore, there are many variations. Among them, the U-net model is one of the most popular fully convolutional network models, and it is widely used in the medical image-processing field. The U-net model is a pixel-to-pixel, end-to-end, fully convolutional network that has skip layers between analysis path and synthesis path. It became popular because of the feature by which it can reserve numerous important features with a small training dataset. Because of the low depth, the U-Net also has a large space for performance improvement by increasing the depth. However, it may lead to gradient vanishing and redundant computation during training while raising the network layers for higher model performance. Gradient vanishing means that the learning rate decreases with forward propagation, which may decrease the overall network learning if the network contains too many hidden layers.

In this paper, based on the U-Net architecture, we proposed an improved method, RCU-Net, to achieve precise tumor segmentation in breast ultrasound images. We added Res Path, which goes from the encoder (left half of the U-Net) to the decoder (right half of the U-Net), and a new connection, a dense unit connection between the input and the decoder, in the proposed method. The Res Path alleviated the differences in the characterization of the feature map between the left-side encoder and the right-side decoder of the U-Net structure. Additionally, the new connection could accelerate the convergence rate of the model, enhancing the feature extraction function and reducing the over-fitting degree of the model. Then, we compared the proposed method with the U-Net and U-Net with just Res Path on a breast ultrasound dataset, which was marked by two professional doctors to verify the effect of our improvements. The results showed that the proposed method had a higher segmentation accuracy than other existing similar methods in terms of true-positive rate, false-positive rate, Hausdorff distance indices, Jaccard similarity, and Dice coefficients. This means that it can reduce misdiagnosis and provide a more accurate diagnosis for doctors.

## 2. Material and Methods

### 2.1. Data Introduction

#### 2.1.1. Data Acquisition

The database was collected from Xinhua Hospital in Shanghai. It contains images obtained using a SAMSUNG RS80A color Doppler ultrasound diagnostic instrument (equipped with a high-frequency probe l3–12a). The images distinctly show the morphology, internal structure, and surrounding tissues of the lesion. All the ultrasound images in the database were of breasts of female patients who were neither pregnant nor lactating. Their ages ranged from 24 to 86. The patients had no history of radiotherapy, chemotherapy, or endocrine therapy.

According to the diagnostic criteria of the breast-imaging reporting and data system (BI-RADS), BI-RADS 2, 3, and 5 exhibit typical benign and malignant characteristics; therefore, the diagnostic accuracy is relatively high. However, for BI-RADS 4, which lack typical benign and malignant characteristics, ultrasonic diagnosis is difficult [13]. In this database, the BI-RADS 4 category of lesions were diagnosed by two medical doctors. For the diagnosis results to be accepted as final, both doctors must agree. If they were not in agreement, a final diagnosis was obtained through mutual consultation.

The database also contained extensive information, including outpatients and tips derived from the ultrasonic information, as well as the corresponding pathological diagnosis. Some patients also underwent the molybdenum target examination and a molybdenum target conclusion was obtained (shown in Figure 1). For the sample images of fibroadenoma (Figure 1a), two pictures were obtained from the same patient. The database shows that patients diagnosed with double milk fibroadenoma generally required mastectomy and pathological diagnosis. The ultrasonic tips revealed double milk hypoechoic and low ultrasonic echo as well as large fibroadenoma (multiple masses in the right breast, Figure 1a above, BI-RADS 3; calcification in the left breast, Figure 1a below, BI-RADS 2); it was necessary to combine ultrasound imaging with clinical practice. Figure 1b shows a set of images from a patient with invasive ductal carcinoma. Figure 1c shows two different patients who were pathologically diagnosed with lobular hyperplasia.

In this study, we selected the images obtained from 256 breast disease patients who were treated between March and October 2013 from the database. The total number of pictures was 538. The maximum diameter of the tumor in these pictures was 3–52 mm.

#### 2.1.2. Image Preprocessing

Figure 1 shows the original ultrasound tumor images we collected. It is obvious that they contain significant interference information that were not conducive to segmentation model training. Therefore, we obtained the images without interference information by manual. Since the ideal input image size for the model was 256 × 256, the original image was scaled to the required size using bicubic interpolation. Bicubic interpolation entails obtaining a new pixel value by summing up the weight convolution of 16 pixels of the image (Equation (1)).
(1)$$p\left(x,y\right)=\overset{3}{\underset{i=0}{\sum }}\overset{3}{\underset{j=0}{\sum }}a_{ij}x^{i}y^{j}$$
where p(x,y) are the interpolation points, and $a_{ij}$ is the weight of the surrounding 16 pixels [14].

Moreover, we observed that different breast ultrasound images in the dataset had different contrast and gray scales. For example, Figure 2b is the gray histogram of Figure 2a, and its gray values focus on the section between 0–150. This makes the feature extraction more difficult for the models. Therefore, we used the Contrast Limited Adaptive Histogram Equalization (CLAHE) to accentuates the contrast between the tumor and surrounding tissue. As the equalization accentuated the contrast, it also distributed the gray histograms in an image that was only concentrated in certain gray intervals over the entire gray interval (Figure 2d). The image after CLAHE is shown in the Figure 2a. It is obvious that the tumor can be observed clearly.

### 2.2. Segmentation Model

#### 2.2.1. U-Net Architecture

U-Net [15] is the combination of a CNN-based encoding-to-decoding process and skip connections. The name of U-Net derives from the U-shaped model. The left part of the model is called encoder according to its function, and the right is called decoder. The contracting path (left side) and an expansive path (right side) are central to its overall framework. The left side encodes for feature extraction, and the right side performs a decoding process for precise positioning. The entire coding process on the left side of the network entails four pooling operations. After each pooling operation, the size of the output image of each layer is doubled together with five sizes, including the original image. Because the output size of the image changes after each sampling, the decoding process on the right side was also sampled to correspond to the encoding process on the left side. By combining the feature map of the encoding part with the corresponding feature map of the decoding part through a shortcut connection, an effective information transmission path was created. This process speeds up the back propagation of the training process and compensates for the low-level details of high-level semantic features.

#### 2.2.2. Residual Path

The U-Net architecture introduces shortcut connections between the encoding layers and corresponding decoding layers. Nabil et al. [16] recently proposed an architecture that incorporates some convolutional layers along the shortcut connections to mitigate differences between the encoding features and decoding features. In addition, because the residual connection simplifies learning, they chose to build the residual connection based on the ordinary convolution layer. The residual unit, Res Path, improves a series of stacking operations directly for the underlying map, alleviates the problem of the vanishing gradient, and connects the feature map from the encoding part to the decoding part.

This method has great potential for application in image analysis. Inspired by this, we introduced Res Path into our segmentation model. Because the receptiveness of the corresponding layer in the encoder stage is different from that in the decoder stage, a series of nonlinear operations can transfer the breast cancer tumor information extracted from the encoder stage to the decoder stage in the training process more effectively. As shown in Figure 3, a series of small residual modules were connected together. Furthermore, 3 × 3 filters were adopted for the convolution layer of each small residual module. Convolutional network residual connections, specifically additional nonlinear operations, were utilized between the corresponding maps of the encoder and decoder, simplifying network learning and reducing the semantic gaps between features.

The residual unit can be expressed as follows:(2)$$x_{l}=H_{l}\left(x_{l-1}\right)+x_{l-1}$$
where $x_{l}$ is the output of the $l^{th}$ layer, $H_{l}$ is a nonlinear transformation to the output of the l−1 layer, the residual block is introduced, and $x_{l}$ is the result that $H_{l}$ sums the identity mapping of the previous layer.

#### 2.2.3. Novel Dense Unit Connection

Dense unit: Based on the idea of the residual block, which combines the identity map of the input with the output of one layer, dense-net [11] builds a more complex connection operation by iteratively connecting feature map output for each previous layer in a feedforward fashion. This connection mode enables the features to be better reused, and the output of all layers is directly supervised by the previous features throughout the unit system. The output of the $l^{th}$ layer is defined as $x_{l}=H_{l}\left(\left[x_{l-1},x_{l-2},\cdots ,x_{0}\right]\right)$, where $\left[x_{l-1},x_{l-2},\cdots ,x_{0}\right]$ denotes the concatenation connection method. Dense blocks with a growth rate k are used to learn the desired number of feature maps, f. The growth rate is updated at each layer so that all the dense blocks have equal number of convolutional layers. In a dense block, earlier convolutional layers are connected to all the subsequent layers through a channel-wise concatenation operation [17].

The output of the $l^{th}$ layer has feature maps, whereas its input has F+k×(l−1) feature maps, where F is the number of feature maps in the input. The hyperparameter k is the growth rate and regulates the number of feature maps learned at each layer.

Inspired by the idea of residual learning, we introduced a new connection that incorporates a dense block from the input of the feature maps in the encoding-to-decoding component. As shown in Figure 4, four concatenation operations form a dense block. As can be observed, the dense block is the skip connection in the residual unit, *X*. For the network, the progress of the inputs first to the encoding section and then to the decoding section is regarded as the big nonlinear transformation *R* = *Y* − *X*. For the encoding-to-decoding part, in the process of network training, one pooling is used, and the size of the image is twice as small, resulting in the loss of image feature information. However, the new connection is a continuous, four-stage convolution operation than can compensate for information loss without a significant loss of information in the process of pooling and up-sampling. As can be seen, in the last stage of decoding, the number of convolution kernels was 128. To keep the same size parameters as the output to fit the add operation, we first selected the two parameters, *f* = 64 and *k* = 16.

#### 2.2.4. Segmentation Architecture

The detailed structure of the improved model is shown in Figure 5. First, we added Res Path to the U-Net framework to reduce the disparity between the encoding and decoding layers to improve the gap between their feature maps in the original shortcut connection and alleviate the problem of gradient vanishing. Because the number of filters for each layer of the U-Net were 64, 128, 256, 512, and 1024, we selected, in turn, 4, 3, 2, and 1 residual units along the four Res Paths. We selected 64, 128, 256, and 512 convolution kernels of the residual unit along the different four Res Paths. We also introduced a dense unit connection between the inputs and the last step of the decoder. We used a dense unit to reduce the gap between the input feature map and decoder feature map. One of the benefits of using the dense block is that a down-sampling operation is not required to move from one layer to the next; thus, fewer features are lost, and more valid information is preserved, which improves the segmentation accuracy.

### 2.3. Training Method and Experimental Design

In this study, 90% of the data were used for training, and the remaining 10% were used for testing. In order to get a reliable and stable model, 10-fold cross-validation was applied in the experiment. A total of 538 breast ultrasound tumor images were randomly divided into 10 cases. For each case, 54 images were used for testing, and the remaining 484 images were used for training. To improve the generalization of the model, we doubled the 484 images through image augmentation, thus obtaining 968 training images.

We used Keras [18] programing language, based on TensorFlow, to implement the models on a Core i7 7700 K processor and 16 GB RAM with a NVIDIA TITAN XP GPU. We utilized the weights ‘he_normal’ [19] to initialize the parameters of the model, making the models converge faster. During the experiment, various kinds of optimization algorithms were tried. The Adam optimization algorithm is a combination of algorithm Momentum and Adagrad, and it had better performance compared with others. Thus, the model was trained using an Adam optimizer [20] with 120 epochs.

The effective step-size adjustment was $\Delta_{t}=\alpha \cdot {\hat{m}}_{t}/\sqrt{{\hat{v}}_{t}}$, where ${\hat{m}}_{t}$ is the exponential moving mean, ${\hat{v}}_{t}$ is the square gradient, ${\hat{m}}_{t}=mt/\left(1-\beta_{1}{}^{t}\right)$, ${\hat{v}}_{t}=v_{t}/\left(1-\beta_{2}{}^{t}\right)$; $\beta_{1}$ and $\beta_{2}$ controlled the decay rate of the moving mean index. The batch size was set to 4, $\beta_{1}$ to 0.9, $\beta_{2}$ to 0.999, and the learning rate α to 0.0001.

In terms of the loss function, because of the binary labels in the segmentation tasks, a cross-entropy function (3) was used as the loss function. For an original input image X, the training label was Y. $\hat{Y}$ was a mask obtained by predicting a trained model. $y_{px}$ and ${\hat{y}}_{px}$ were the ground truth and the result of the test for a pixel in an image. The cross-entropy loss became
(3)$$Cross\;Entropy\left(X,Y,\hat{Y}\right)=\underset{px\in X}{\sum }-\left(y_{px}\log\left({\hat{y}}_{px}\right)+\left(1-y_{px}\right)\log\left(1-{\hat{y}}_{px}\right)\right)$$

For a batch including n ultrasound images, the loss function L is defined as follows:(4)$$L=\frac{1}{n}\overset{n}{\underset{i=1}{\sum }}Cross\;Entropy\left(X_{i},Y_{i},{\hat{Y}}_{i}\right)$$

Under the same condition mentioned above, the three models (U-Net, U-Net with Res Path, and the proposed model) were trained using 10-fold cross validation. Each train used the same training and test samples. The classic U-Net model was used as the reference to verify the validity of our improvements. The model—U-Net with Res Path—was derived by incorporating the Res Path into the basic network structure of the U-Net to alleviate the problem of the gradient vanishing and the disparity between the encoding and decoding sections. It was introduced into the comparison to prove that it is the new connection that brought about the improvements rather than the Res Path. The proposed model was used to show the improvement.

### 2.4. Evaluation Metric

For quantitative performance evaluation, the experimental results were evaluated using five performance metrics as follows: true positive (TP), false positive (FP), Dice coefficient (DC) [21], Jaccard similarity (JS) [22], and Hausdorff distance (HD). In order to define these measures, we also used the variables Ground Truth (GT) and Segmentation Result (SR). The GT represents the segmented region, and all the imaging datasets were segmented manually by professional doctors following the standard annotation protocol. These GT contours are references for further segmentation analysis [23]. The SR represents the segmentation result from the method to be evaluated.

The TP is the proportion of pixels predicted correct in the segmentation results to the GT (Equation (5)). The FP refers to the proportion of the pixels predicted wrong in the segmentation results to the GT (Equation (6)). The higher the TP value, the greater the coverage of the target region; the lower the FP value, the fewer background areas containing misclassification.
(5)$$\mathrm{TP}=\frac{\left|\mathrm{GT}\cap \mathrm{SR}\right|}{\left|\mathrm{GT}\right|}$$
(6)$$\mathrm{FP}=\frac{\left|\mathrm{GT}\cup \mathrm{SR}-\mathrm{GT}\right|}{\left|\mathrm{GT}\right|}$$

The DC is an index used in statistics to measure the accuracy of the binary classification model. The closer DC is to 1, the more accurate the segmentation result will be. It can be calculated using Equation (7):(7)$$\mathrm{DC}=2\frac{\left|\mathrm{GT}\cap \mathrm{SR}\right|}{\left|\mathrm{GT}\right|+\left|\mathrm{SR}\right|}$$

The JS is used to compare the similarities and differences between finite sample sets. Its value is directly proportional to the similarity (Equation (8)):(8)$$\mathrm{JS}=\frac{\left|\mathrm{GT}\cap \mathrm{SR}\right|}{\left|\mathrm{GT}\cup \mathrm{SR}\right|}$$

The HD is the maximum of all the shortest distances between two points. For ultrasound images, HD refers to the distance between each point in the predicted results and the point on the standard mask (Equation (9)).
(9)$$\mathrm{HD}\left(X,Y\right)=max\left\{\begin{array}{c} \sup\;\inf\;d\left(x,y\right),\sup\;\inf\;d\left(x,y\right)\\ x\in X,y\in Y\;y\in Y,x\in X \end{array}\right\}$$
where the sup and inf represent the upper and lower bounds, respectively. *X* and *Y* represent the tested label and corresponding ground truth, respectively.

## 3. Results

For each model, 10 sets of weights were generated using 10-fold cross validation. The testing images were input into the models with different weights, respectively. Then the output images were used, combined with corresponding label, to evaluate the performance of models by comparing the five metrics. Table 1 summarizes the mean and standard deviation of the five metrics.

It can be observed that the performance of U-Net was significantly improved after incorporating the Res Path, based on all four indicators apart from the HD. Furthermore, the proposed model not only surpassed the original U-Net but also outperformed the U-Net with Res Path. In terms of the five metrics, the proposed model was superior to the others in different degrees. Compared with that of the U-Net, the TP, JS, and DC of the proposed model improved by 1.0625%, 2.135%, and 2.0105%, respectively. The HD index decreased by 1.41023.

We selected four typical ultrasound images of breast cancer with different edge contours, tissue texture features, and contrasts (Figure 6a). And the outputs of different models are shown on the right. The testing results indicates that the three models could complete the segmentation task effectively when the tumor has a defined shape, such as the first, third, and fourth row in the Figure 6a. However, in the case that the tumor has a fuzzy boundary, such as the second row in Figure 6a, there were many pixels that were wrongly predicted in the output of the original U-Net model. In addition, the U-Net with Res Path had many fuzzy pixels on the edge of segmentation position. It was the Res Path that greatly preserved the spatial structure of the gradient, making the output of the model stable. Because the dense block in the proposed model enhanced the feature transmission between the encoder and decoder, the result shows better performance compared with the other models.

Figure 7 shows the bar plots of the 10 cases for the three models. As depicted, the proposed method had higher TP and lower FP in most cases. It also had more significant excess than the others in a few cases. In the terms of TP, the proposed model had a more stable performance than U-Net with Res Path, though they had close scores in most cases. However, in view of the FP, the proposed model was overwhelming.

In general, a segmentation was considered as successful when JS > 0.75. Therefore, the number of images in which JS > 0.75 can directly show the performance of segmentation. The box plots in Figure 8 shows the statistics of average values in each case of 10-fold cross-validation. As the green lines show, the median of the proposed model was the highest among the models. The U-Net with Res Path even performed poorer than the U-Net in median and third quartile but had a lower deviation. But in the proposed model, it significantly increased the median while keeping the advantage in deviation. This means that the proposed model had a higher segmentation accuracy in most cases than the others.

## 4. Discussion

In this paper, aimed at the problem that original U-Net model performed poorly with a low segmentation accuracy when used in the auxiliary diagnostic system for breast tumor segmentation, we proposed an improved model based on U-Net. The main reason for the low performance is the large semantic differences and the vanishing gradient problem of the U-Net. Therefore, we introduced the Res Path into the U-Net to reduce the feature maps difference between the encoder and decoder. Then, a new connection, dense block from the input of the feature maps in the encoding-to-decoding section, was added to reduce the feature information loss and alleviate the vanishing gradient problem. To verify the improvements, we used a breast tumor ultrasound dataset marked by professional doctors to train and test three models—the original U-Net, U-Net with only Res Path, and the proposed model.

As seen from the models output Figure 6, the U-Net had lesser number of pixels false predicted due to Res Path. However, the Res Path also caused the problem that the segmented tumor is internally incomplete while reducing feature maps difference between the encoder and decoder. Low-level features from the decoder have large receptive fields, which in turn cause internal prediction errors. Therefore, we introduced the dense block to address this issue. The new connection containing dense block is a continuous, four-stage convolution operation that can compensate for information loss without a significant information loss in the process of pooling and up-sampling. It helped reduced the internal incompleteness of Res Path. The results in the Figure 8 show this point well, while the statistics show the average performance.

To show the performance of our proposed model, we compared some models used in medical image segmentation. We trained them in our dataset and maximized their performance as much as possible. Under the same conditions, we tested their performance using the metrics mentioned above. Because the workload was huge if using 10-fold cross-validation, we employed only five-fold cross-validation in the experiment. Table 2 summarizes the comparison results. The results show that the U-Net, DIU-Net, and the proposed model show better performance compared with SegNet and FCN-8s because of the skip connection. Additionally, the DIU-Net had close performance to our proposed model due to the deeper layer. However, excessive layers may lead to too many parameters, making the model more difficult and slower to train. Moreover, given the hospital computer performance, it is unreasonable to make the model too complex.

With the help of CAD, diagnosis speed and medical cost has been declining for the past few years. However, it may cause an opposite effect if the segmentation accuracy is insufficient. The problem of disappearing gradient and excessive semantic differences are common obstacles for improving the performance of neural network models. The introduction of using dense block to alleviate the semantic gap between different levels can provide help for relevant studies. The improvement in accuracy may not be dramatic, but it is also considerable when the number of patients increases.

There are some limitations in this study, though the model performed better than others. There were still some failed segmentation cases owing to the difficulty of extracting features in some images. As shown in Figure 9, the boundary of the tumor was too obscure to extract the features. Another main reason for this is the number of the cases in which the tumor’s boundary is extremely obscure was very few. Hence, the model could not acquire enough experience. Therefore, a complete and manifold dataset are necessary for a high-accuracy model.

The model proposed in this study is an improvement on the original U-Net model. After the introduction of dense blocks and residual paths, the parameters increased significantly, and the computational memory became larger, which necessitates better hardware support. We will aim to simplify the network structure whilst maintaining its accuracy and effectiveness in the next work. Another issue is that we only tested the model in our private breast tumor ultrasound images. Images of other types of tumors or a miniature dataset may lead to decreased performance. Therefore, we will test the proposed model on various dataset in the future and modify it to make the model more effective on various kinds of medical problems.

## 5. Conclusions

Introducing Res Path into U-Net can effectively reduce the vanishing gradient problem, but it also brought about prediction error inside the segmentation result. The novel connection composed of dense blocks can address this problem and at the same time reduce the semantic difference between the encoder and decoder. The model using both two modules can result in a significant improvement in segmentation accuracy on breast tumor ultrasound images.

## Figures and Tables

**Figure 1 diagnostics-11-01565-f001:**
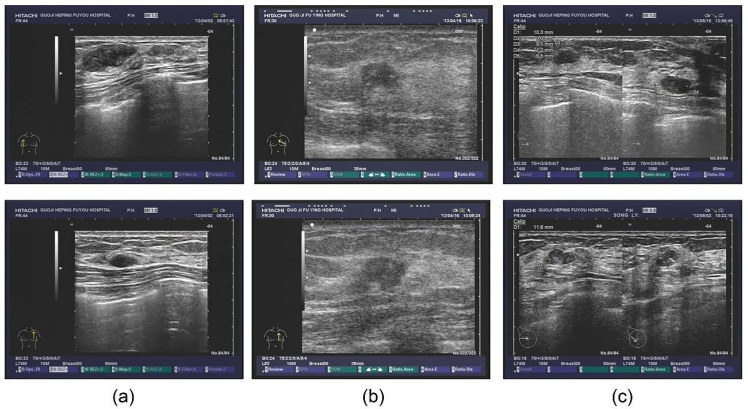
Original ultrasound images. (**a**) examples of fibroadenoma; (**b**) examples of invasive ductal carcinoma; (**c**) examples of lobular hyperplasia.

**Figure 2 diagnostics-11-01565-f002:**
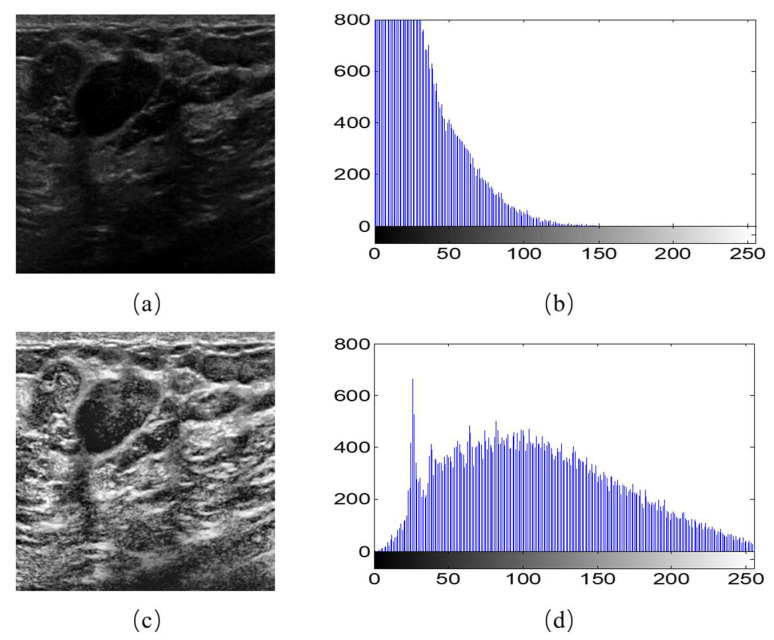
Image histogram equalization. (**a**) An example of original image; (**b**) gray histogram of (**a**); (**c**) the image after histogram equalization; (**d**) gray histogram of (**c**).

**Figure 3 diagnostics-11-01565-f003:**
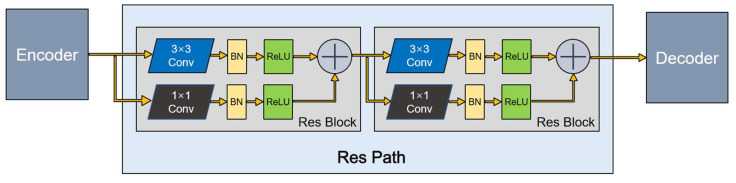
A Res Path with only two Res Block.

**Figure 4 diagnostics-11-01565-f004:**
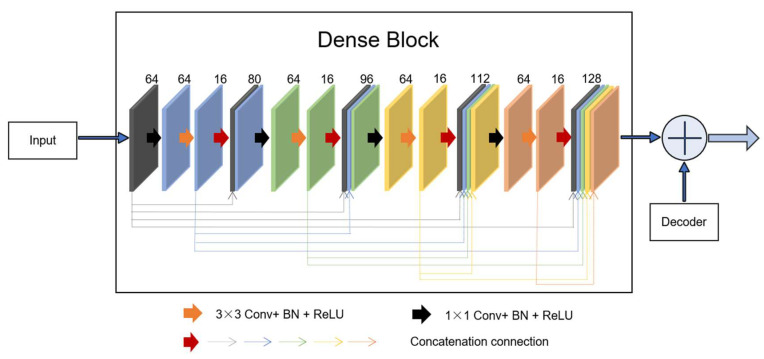
A new connection composed of dense block. In the dense unit, feature-maps learned in previous layers are concatenated with subsequent layers, finally to the desired *f* = 128 features.

**Figure 5 diagnostics-11-01565-f005:**
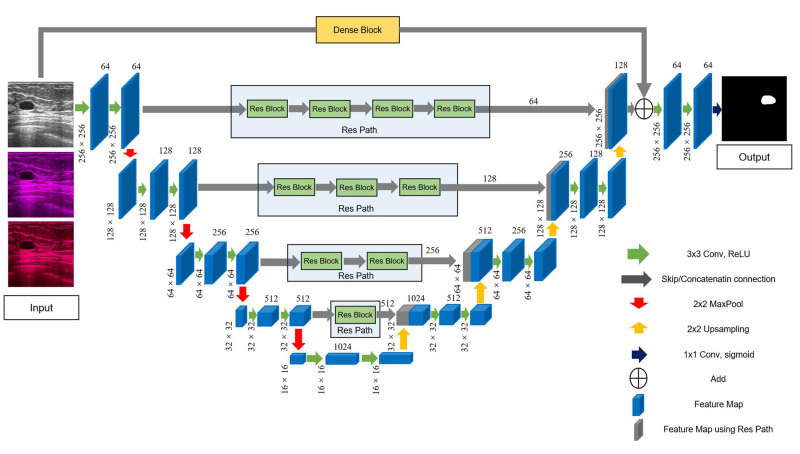
The main framework of the proposed model, RCU-Net. The number of convolution cores in each Res Path corresponds to the number of convolution cores in the previous layer of encoder.

**Figure 6 diagnostics-11-01565-f006:**
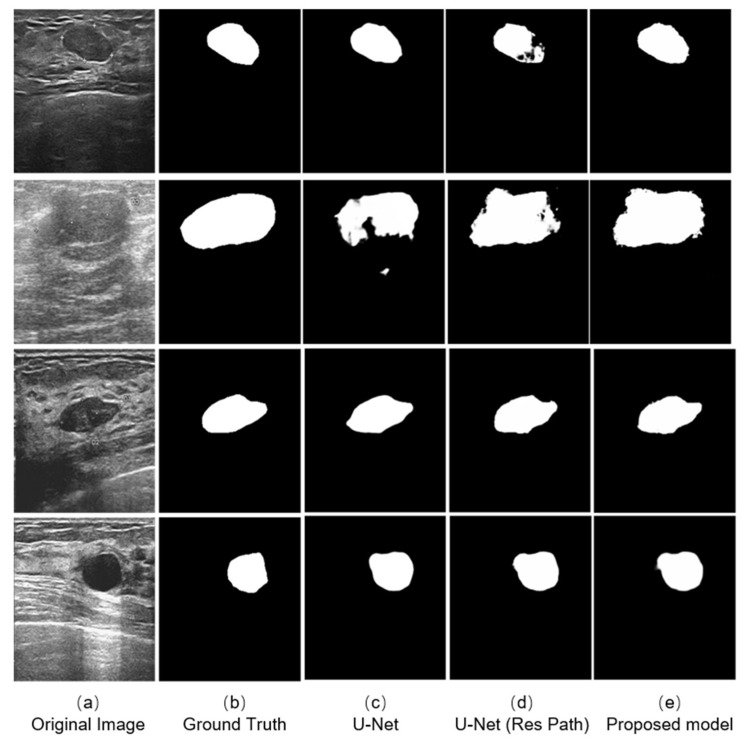
Experimental outputs using different kinds of methods are: (**a**) the ultrasound images; (**b**) the labeled figures marked by the doctor; (**c**) the outputs of the U-Net model; (**d**) the outputs of the U-Net with Res Path Model; and (**e**) the outputs of our proposed model.

**Figure 7 diagnostics-11-01565-f007:**
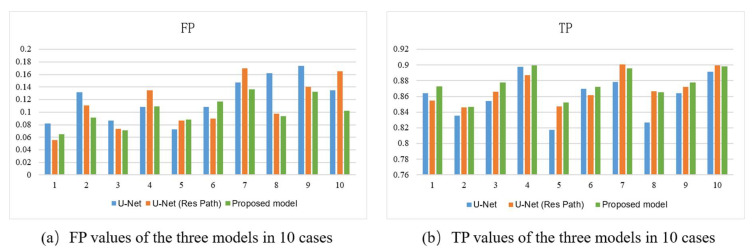
(**a**) FP, (**b**) TP of the outputs of models using 10-fold cross-validation. The *x*-axis represents the case number and the *y*-axis the average value of the metric.

**Figure 8 diagnostics-11-01565-f008:**
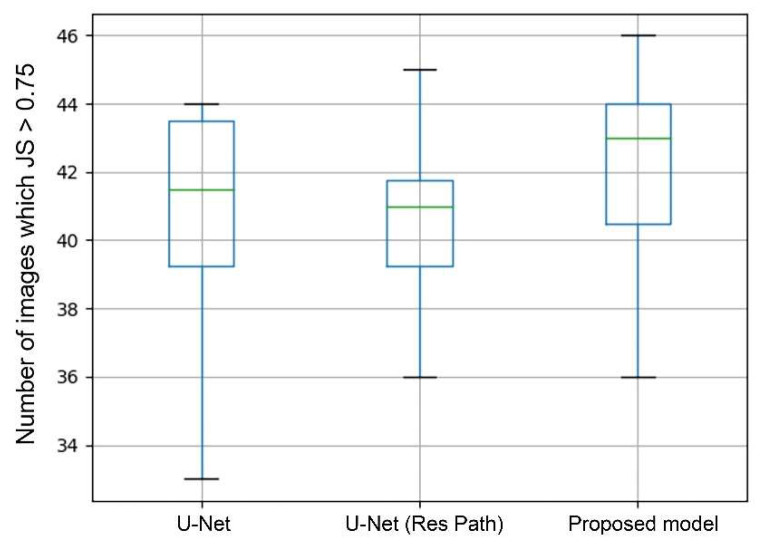
Box plots of the image number of JS > 0.75. The green line represents the median. The upper black horizontal line represents the max value and the lower min value. Top of box denotes the 3rd quartile and the bottom 1st quartile.

**Figure 9 diagnostics-11-01565-f009:**
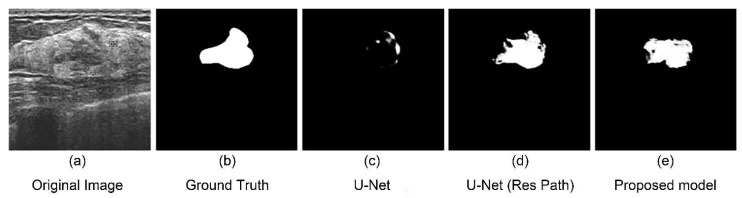
An unsuccessful segmentation case. (**a**) Original figure; (**b**) Ground Truth obtained from (**a**); (**c**) segmentation result of the U-Net model; (**d**) segmentation result of the U-Net with Res Path model; (**e**) segmentation result of the proposed model.

**Table 1 diagnostics-11-01565-t001:** The mean and standard deviation of the five indexes using 10-fold cross validation. The numbers behind the “±” are the standard deviation.

	Metric	DC	HD	JS	FP	TP
Model						
U-Net		0.8623 ± 0.0198	23.7216 ± 2.946	0.7823 ± 0.0229	0.1208 ± 0.0291	0.8599 ± 0.0212
U-Net with Res Path		0.874 ± 0.0129	24.4506 ± 1.812	0.7937 ± 0.0135	0.1123 ± 0.0322	0.8705 ± 0.0164
The proposed model		0.8824 ± 0.01108	22.3114 ± 1.4298	0.8037 ± 0.0126	0.104 ± 0.0219	0.8707 ± 0.0131

**Table 2 diagnostics-11-01565-t002:** Comparison of different segmentation models.

	Metric	DC	HD	JS	FP	TP
Model						
SegNet [24]		0.8602	23.7365	0.7742	0.1425	0.8421
FCN-8s [12]		0.8616	23.8264	0.7765	0.1386	0.8432
U-Net		0.8623	23.7216	0.7823	0.1208	0.8599
DIU-Net [25]		0.8795	22.3274	0.8035	0.109	0.8702
The proposed model		0.8824	22.3114	0.8037	0.104	0.8707

## Data Availability

The data we used was private dataset.

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
