# Peer review of "Tumor Segmentation in Breast Ultrasound Image by Means of Res Path Combined with Dense Connection Neural Network"

_diagnostics, 2021, doi:10.3390/diagnostics11091565_

Round 1

Reviewer 1 Report

Thank you for acknowledging all my comments and concerns. The manuscript has been significantly improved.

Reviewer 2 Report

The resubmitted manuscript represents a better version, with all the corrections required. Moreover, the Authors improved the introduction and the discussion, explaining in a more understandable way the background, the methods and the results of their study and increasing the possible interest to the readers.

This manuscript is a resubmission of an earlier submission. The following is a list of the peer review reports and author responses from that submission.

Round 1

Reviewer 1 Report

In this interesting study the authors proposed a new deep learning model to improve the precision of tumor segmentation in breast ultrasound imaging. The results showed moderate, but statistically valid, improvements in terms of true positive and false positive rates, when compared with other deep learning segmentation models. I would suggest, if authors agree, to improve the discussion section, explaining in a more easily understandable way the results and highlighting the relevance of this new method, in order to enlarge the interest to the readers.

Reviewer 2 Report

The authors have presented the work titled " Tumor Segmentation in Breast Ultrasound Image by Means of Res Path Combined with Dense Connection Neural Network", and they proposed a method to improve the precision of segmentation using the improved classic deep learning segmentation model (U-Net). Here are few comments to improve the quality of this work: 

  1. In lines 21-23, this sentence should be moved to at the beginning or in the middle of the abstract.
  2. As this work has relation with breast tumors, so it is advised to improve the picture quality of Figure 1 and Figure 2. 
  3. Include the reference for equation 1. 
  4. In line 195, the authors talked about the left side and right side, It's talking about figure 2, please elaborate. 
  5. In line 170, the authors used "Figure" and the rest of the paper "Fig", make it the same. 
  6. Figure 4 is not cited in the text body. 
  7. The caption of Figure 5 is too large, make it small, and also check for other figures as well. 
  8. In line 283, the authors talked about experiments, it is necessary to include the picture of the experimental setup. 
  9. Use some other colors to make Figure 7 more bright. 
  10. In table 3, the authors have compared their proposed work with some generic difference techniques. As this work is an improved form of U.Net so better to compare this work related to its work (Reference: 26-30).

Reviewer 3 Report

The manuscript by Chen S and Chen Y represents an effort to describe a new method for tumor segmentation using breast ultrasound images. While the manuscript certainly has got some interesting parts, it fails to convince me about the importance of the research done. 

Comments:

The authors must improve the introduction, and differentiate the imaging techniques, explaining in which situations the ultrasound imaging is done. The ultrasound is not a screening method for breast cancer, therefore we have to know when is it done, and why is it important. And this is in direct correlation with the significance of this manuscripts research findings. Which patients can benefit from the method described by authors? The authors state that their findings may spur further research in different areas, but they fail to explain how exactly and also how will that benefit the patients.

Authors must improve the writing, as the results and discussion parts are written in a very confusing manner, and it was hard to follow. Figures and tables should be better labeled perhaps, with less explanation in the figure description and more explanation in the text. Figure 6 should be better explained in the text.

While I can definitely recognize the effort, the manuscript lacks novelty and showing the importance of the problem studied.